# Clinical Utility of Leeds Dependence Questionnaire in Medication-Overuse Headache

**DOI:** 10.3390/diagnostics13030472

**Published:** 2023-01-27

**Authors:** Yen-Feng Wang, Yi-Shiang Tzeng, Chia-Chun Yu, Yu-Hsiang Ling, Shih-Pin Chen, Kuan-Lin Lai, Shuu-Jiun Wang

**Affiliations:** 1Department of Neurology, Neurological Institute, Taipei Veterans General Hospital, Taipei 11217, Taiwan; 2School of Medicine, National Yang Ming Chiao Tung University, Taipei 11217, Taiwan; 3Brain Research Center, National Yang Ming Chiao Tung University, Taipei 11217, Taiwan; 4Division of Translational Research, Department of Medical Research, Taipei Veterans General Hospital, Taipei 11217, Taiwan

**Keywords:** chronic migraine, medication-overuse headache, dependence, questionnaire

## Abstract

Dependence behaviors are common in patients with medication-overuse headache (MOH). This prospective study aimed to characterize dependence behaviors in MOH by using Leeds dependence questionnaire (LDQ), and to determine the clinical utility of LDQ in the diagnosis of MOH. In total, 563 consecutive chronic migraine (CM) patients (451F/112M, mean age 41.7 ± 12.0 years) were recruited, including 320 with MOH (56.8%) (254F/66M, mean age 42.3 ± 11.6 years). LDQ scores were positively correlated with the monthly frequency of acute medication use (Spearman’s rho = 0.680, *p* < 0.001). When compared with patients without, those with MOH scored higher on LDQ (13.0 ± 7.6 vs. 3.9 ± 5.1, *p* < 0.001). By using a receiver operating characteristics curve, the cutoff value of LDQ was determined at 7 (sensitivity = 77.5%, specificity = 77.4%, area under curve = 0.85) for a diagnosis of MOH. An LDQ score of ≥7 was predictive of MOH (odds ratio = 11.80, 95% confidence interval = 7.87–17.67, *p* < 0.001). In conclusion, the presence of MOH in patients with CM is associated with more severe dependence behaviors. An LDQ score of ≥7 is useful in the detection of MOH in CM patients.

## 1. Introduction

Medication-overuse headache (MOH) is characterized by a paradoxical increase in headache frequency and severity following regular overuse of acute medications for a long period of time [1]. MOH typically evolves from a pre-existing primary headache disorder [2]. Withdrawal of the overused acute medications may lead to headache improvement and gradual reversion to the underlying primary headache disorder [1]. The prevalence of MOH is 0.5–2.6% in the general population [3], which is consistent in different parts of the world, including Taiwan [4,5,6]. MOH is frequently seen in the middle adulthood [7], which consists of the most productive years. It is the third most prevalent, and the sixth leading cause of disease-related disability among all neurological disorders, according to Global Burden of Disease Study 2015 by the World Health Organization [8]. Despite the tremendous impact on the quality of the life of patients suffering from MOH and the consequent enormous socioeconomic burden [3,8], this condition remains under-recognized and under-treated.

The majority of MOH patients have migraine as the underlying primary headache disorder [9]. Migraine is associated with an increased risk for developing psychiatric comorbidities, such as depression, anxiety disorders, etc., which are also believed to be involved in migraine chronification and development of MOH [10,11]. Interestingly, it was also reported that about two thirds of MOH patients could fulfill the criteria for substance dependence in the fourth edition of Diagnostic and Statistical Manual of Mental Disorders (DSM-IV) [12,13,14]. Clinically, misuse or even abuse of psychoactive substances other than analgesics was reported to be present in a significant proportion of MOH patients or even in their family [15]. Supportive evidence could be derived from neuroimaging studies. It was found that MOH patients had persistent hypometabolism in the orbitofrontal cortex (OFC) on fluorodeoxyglucose positron emission tomography, which persisted after detoxification [16]. Furthermore, CM patients with MO had decreased gray matter volume in the OFC, which was predictive of treatment response [17]. In fact, structural or functional alterations in the OFC are important features of substance use disorders, and are related to drive, compulsive repetitive behaviors, or even relapse [18,19]. Therefore, MOH seems to bear resemblance to substance use disorders from clinical and pathophysiological perspectives. The above findings are suggestive of the presence of impairment in the reward circuit in MOH patients, as seen in those with substance use disorders. However, more clinical evidence supporting such a hypothesis is needed.

Dependence behaviors in MOH are gaining increasing attention is recent years and are of clinical interest and importance for diagnostic and prognostic purposes. It was found that chronic daily headache (CDH) patients with medication overuse (MO) had more severe dependence behaviors compared with patients with episodic migraine, episodic cluster headache, or episodic tension-type headache (TTH) [20]. Moreover, it was demonstrated that the severity of dependence behaviors was not only correlated with medication overuse (MO) in the general population [21] but was also predictive of the prognosis of detoxification in primary MOH [22,23]. However, many prior studies involved comparisons between MOH patients and patients with episodic headache disorders or even healthy controls, and data on direct comparisons between CM patients with and without MOH are scarce. Furthermore, the Severity of Dependence Scale appeared to be more widely used in the studies of MOH. Whether the Leeds Dependence Questionnaire (LDQ), another neuropsychological instrument commonly used in the measurement of dependence behaviors in patients with substance use disorders [24], could also be useful in MOH patients remains to be elucidated.

The present study intended to characterize behaviors of dependence in CM patients with and without coexisting MOH by using the LDQ and to determine the potential use of the LDQ in the screening of MOH in patients with CM.

## 2. Methods

### 2.1. Patients

This was a prospective study involving newly diagnosed CM patients with and without a concomitant diagnosis of MOH. Patients were enrolled consecutively at their first visit to the Headache Clinic of Taipei Veterans General Hospital, a tertiary medical center for both veterans and civilians in Taiwan. The initial evaluation consisted of questionnaire-based interviews by headache specialists. The diagnoses of CM and MOH were made according to the diagnostic criteria of the Third Edition of the International Classification of Headache Disorders (ICHD) (ICHD-3) [25]. Patients were included if they (a) were willing to participate in the study, (b) were aged between 20 and 65 years, and (c) fulfilled the ICHD-3 criteria for CM. The exclusion criteria included (a) an acute headache disorder (within one month of headache onset), (b) a secondary headache disorder, and (c) difficulties completing the history taking or the questionnaire-based interview. The study protocols were approved by the Institutional Review Board of Taipei Veterans General Hospital (protocol number 2018-07-020BC, date of approval: 25 July 2018; protocol number 2019-07-002CC, date of approval: 12 July 2019). All of the patients gave written informed consent before entering the study.

### 2.2. Questionnaire-Based Interviews

The questionnaire was designed to collect demographics, clinical profiles, and headache characteristics, as well as to screen for psychological disturbances and behaviors of dependence. Headache-related disabilities were assessed with the Migraine Disability Assessment Scale (MIDAS) [26]. Symptoms of depression and anxiety were assessed by using the Hospital Anxiety and Depression Scale (HADS), including anxiety (HADS-A) and depression subscales (HADS-D) [27]. The Pittsburg Sleep Quality Index (PSQI) was used to evaluate the severity of sleep disturbances [28]. Dependence behaviors were screened by using the LDQ [29], with some modifications for the use in headache disorders [20]. The scores of these instruments were verified during the face-to-face interviews by headache specialists.

The modified version of LDQ consists of ten questions [20], each which is to be rated on a scale of 0 to 3 (0 = never, 1 = sometimes, 2 = often, 3 = nearly always) (Table 1). The total score ranges from 0 to 30. The questions are as follows: 1. Do you find yourself thinking about when you will next be able to take analgesics? 2. Is taking analgesics more important than anything else you might do during the day? 3. Do you feel your need for analgesics is too strong to control? 4. Do you plan your days around taking analgesics? 5. Do you take analgesic in a particular way in order to increase the effect it gives you? 6. Do you take analgesics morning, afternoon and evening? 7. Do you feel you have to carry on taking analgesics once you have started? 8. Is getting the effect you want more important than the particular analgesic you use? 9. Do you want to take more analgesics when the effect starts to wear off? 10. Do you find it difficult to cope with life without analgesics? This questionnaire was translated into traditional Chinese for its use in Taiwan following standard protocols for cross-cultural research: translation, back translation, and bilingual expert panel evaluation.

### 2.3. Statistical Analysis

Data were expressed as mean ± standard deviations (SD) or number (*n*) (percentage). Continuous variables were compared by using Student’s *t* test, or Mann–Whitney *U* test for non-normally distributed variables. Categorical variables were compared by using chi-square test. Correlations between LDQ scores and clinical parameters, including age, CM duration, frequencies of headache and analgesic use, etc., were evaluated by Spearman’s rank correlation coefficient, as many of the data were not normally distributed. The optimum cut-off score of the LDS for detecting patients with coexisting MOH was determined by using a receiver operating characteristics (ROC) curve, in conjunction with Youden’s J statistics. The areas under the ROC curves (AUC) were calculated to evaluate the discriminative performance of these instruments. In general, an AUC of 0.7 to 0.8 indicates acceptable, 0.8 to 0.9 excellent, and >0.9 outstanding accuracy of a diagnostic test [29]. Logistic regression modeling was carried out to examine the association between the cut-off score of the LDQ and the diagnosis of MOH, and to estimate the odds ratios (ORs) and the 95% confidence intervals (CIs). Statistical analysis was carried out by using IBM SPSS Statistics for Windows, version 24.0 (IBM Corp., Armonk, NY, USA). Statistical significance was defined as a two-sided *p* of <0.05.

## 3. Results

### 3.1. Demographics and Clinical Characteristics

In total, 563 consecutive CM patients (451F/112M, mean age 41.7 ± 12.0 years) were recruited, including 320 with MOH (56.8%) (254F/66M, mean age 42.3 ± 11.6 years) (Table 1). When compared with patients without MOH, those with MOH were less likely to have a bachelor’s degree or higher (43.4% vs. 56.0%, *p* = 0.003), although they had a longer duration of CM (90.4 ± 100.6 vs. 60.4 ± 112.4 months, *p* < 0.001), greater average intensity (7.0 ± 1.9 vs. 6.2 ± 1.8, *p* < 0.001), many more days per month with acute medication use (19.4 ± 7.8 vs. 3.7 ± 5.3 days/month, *p* < 0.001), greater disabilities (MIDAS 66.2 ± 77.4 vs. 42.5 ± 51.3, *p* < 0.001), and poorer sleep quality (PSQI 11.9 ± 4.4 vs. 11.1 ± 4.2, *p* = 0.030). Furthermore, there was a trend toward an earlier onset of migraine in patients with MOH (20.7 ± 9.2 vs. 22.8 ± 10.8, *p* = 0.065). On the other hand, the age, gender distribution, marital status, employment status, the presence of migraine aura, the number of monthly headache days, and scores on the HADS-A, HADS-D and HADS-T were comparable between these two groups.

### 3.2. Dependence Behavior

Among the entire study population, the LDQ score was positively correlated with the number of days per month with acute medication use (Spearman’s rho = 0.680, *p* < 0.001) and CM duration (Spearman’s rho = 0.304, *p* < 0.001), although there was only a weak correlation with the number of MHDs (Spearman’s rho = 0.095, *p* = 0.024). Moreover, there were also correlations between LDQ scores and the scores of MIDAS (Spearman’s rho = 0.260, *p* < 0.001), PSQI (Spearman’s rho = 0.256, *p* < 0.001), HADS-D (Spearman’s rho = 0.193, *p* < 0.001), and HADS-A (Spearman’s rho = 0.120, *p* = 0.004).

When compared with patients without, those with MOH scored higher in the LDQ (13.0 ± 7.6 vs. 3.9 ± 5.1, *p* < 0.001) (Table 2). In fact, the between-group difference was significant for every single question included in the instrument. The mode of the LDQ for patients with MOH was 10 (Figure 1), which was reported by 6.6% of patients, whereas it was 0 for those without MOH, which was reported by 34.2%.

### 3.3. Clinical Utility of LDQ

By using the ROC curve, the cut-off score of LDQ for a diagnosis of MOH was determined at 7, with a sensitivity of 77.5% and a specificity of 77.4% (Youden’s *J* index = 0.55) (Figure 2), which belonged to the category of excellent diagnostic accuracy (AUC = 0.85 [Asymptotic 95% Confidence Interval = 0.82–0.88]). However, a cut-off score of 4 (sensitivity = 90.3%, specificity = 61.7%) or 5 (sensitivity = 87.2%, specificity = 67.1%) could be more appropriate for screening purposes because of higher sensitivities.

In the entire study population (*n* = 563), an LDQ score of ≥7 was associated with the presence of MOH (OR = 11.77, 95% CI = 7.90–17.55, *p* < 0.001), and the findings were consistent (OR = 11.80, 95% CI = 7.87–17.67, *p* < 0.001) after controlling for demographics (age, sex, education level, marital status, and employment status). In patients without MOH (*n* = 243), an LDQ score of ≥7 was associated with more days per month with acute medication use (6.6 ± 7.4 vs. 2.9 ± 4.1 days/month, *p* < 0.001) and greater disabilities (MIDAS 52.4 ± 42.9 vs. 39.6 ± 53.2, *p* < 0.001). The findings were similar in patients with MOH (*n* = 320) (Monthly analgesic use 20.4 ± 7.6 vs. 15.7 ± 7.5, *p* = 0.002; MIDAS 71.4 ± 82.0 vs. 48.3 ± 55.8, *p* = 0.044) (Figure 3).

## 4. Discussion

In the current study, it was found that CM patients with coexisting MOH had a lower education level, a longer duration of CM, greater headache intensities and disabilities, and poorer sleep quality when compared with those without. More importantly, the presence of MOH was associated with more severe dependence behaviors, as evidenced by higher scores on the LDQ, and the LDQ score was highly correlated with the frequency of acute medication use. Furthermore, a cut-off score of LDQ ≥ 7 was useful in the diagnosis of MOH among CM patients, and it was 4 or 5 for screening purposes. An LDQ score of ≥7 was associated with an increased risk of MOH by 10 folds and was associated with more acute medication use and greater disabilities regardless of the presence of MOH. The LDQ appeared to be a powerful instrument in the screening of dependence behaviors in CM patients. Moreover, a cut-off score of ≥7 on the LDQ should alert the treating physician of the possibility of coexisting MOH. However, whether the treatment response to preventive medications could be different needs to be further studied.

One of the most important strengths of the current study was the sample size. More than 550 CM patients with MO were recruited consecutively, which could be one of the largest studies carried out in the clinical setting so far. A large sample size could reduce selection bias and could give more accurate estimates. Second, direct comparisons were made between CM patients with and without MOH. In comparison, many prior reports compared MOH with episodic migraine, cluster headache, TTH, or even healthy controls [20,21,23,30], and it is uncertain whether these findings were pertinent to MOH or coexisting headache disorders. Therefore, the findings of the present study are more relevant to MOH rather than the underlying CM. Third, the present study was carried out in the clinical setting and involved CM patients. Even though there were studies evaluating the potential use of the Severity of Dependence Scale as a screening instrument for MOH, these studies were population-based and chronic TTH was the predominant headache diagnosis [21,30]. Therefore, the findings of the current study could have more practical impacts for clinicians treating headache patients.

Some of the clinical manifestations of substance use disorders are shared by MOH patients, particularly dependence behaviors [20,21]. It was reported that the need for analgesics, as measured by the LDQ, in CDH patients with daily use of acute medications was comparable to that for drugs of abuse in substance use disorders involving heroin, cocaine, alcohol, cannabis derivatives, and amphetamine [20]. In the current study, the severity of dependence behaviors, as measured by LDQ scores, was strongly correlated with monthly acute medication use. The finding was in keeping with prior reports and suggested the LDQ could be useful in measuring the severity of dependence behaviors in MOH patients. Moreover, the LDQ was useful in the diagnosis or even screening of MOH in CM patients. In the literature, there was a screening tool consisting of two questions, namely “do you take a treatment for attacks more than 10 days per month” and “is this intake on a regular basis” [31], and it was concluded that the tool was sensitive and specific for MOH based on revised criteria of the Second Edition of ICHD [32]. However, it provided only qualitative rather than quantitative information, and could not reflect the severity of the condition. On the other hand, in prior studies, it was also demonstrated that a score of ≥5 on the Severity of Dependence Scale was useful not only in the screening of MO in patients with primary chronic headaches, but also in detecting dependence defined by DSM-IV among MOH patients [14,21,23]. Although the LDQ and the Severity of Dependence Scale were both developed as measures for dependence behaviors [24,33], the LDQ seemed to be under-utilized in clinical studies of headache disorders. Further studies involving direct comparisons on the diagnostic or screening performances of these two instruments would help clarify the role of LDQ in the clinical evaluation and management of MOH.

The severity of dependence behaviors could be an important factor associated with the prognosis of MOH patients. However, whether the LDQ could be predictive of treatment outcome remains to be explored. It was shown that patients with more severe dependence behaviors, as measured by the Severity of Dependence Scale, had a poorer prognosis of detoxification [22], which highlights the importance of screening for dependence behaviors for patients with CDH, including CM. In fact, MOH patients who scored ≥7 on the LDQ in the present study had more frequent acute medication use, which could a risk factor for migraine chronification and relapse of MOH after withdrawal of acute medications [34,35]. However, as a cross-sectional study, the question whether higher LDQ scores could be associated with a less favorable outcome or a higher relapse rate after detoxification could not be answered. Interestingly, it was also found that even in patients without MOH, an LDQ score of ≥7 was associated with more days per month with acute medication use and greater disabilities, when compared with an LDQ score of <7. Whether such patients are at an increased risk of developing MOH in the future deserves further study. 

There were some limitations. In particular, the external and internal validity of the findings of the present study could be important concerns. First of all, the study participants were recruited from the headache clinic of a tertiary medical center, which could potentially limit the generalizability of the findings. However, the present study involved a relatively large sample size, which could potentially reduce selection bias. Furthermore, a formal referral system is yet to be developed in our country, and most of patients came in directly without referral. Second, although the prospective nature reduced the risk of recall bias, the cross-sectional design could only give information for an association. Further studies of a longitudinal design are needed to determine whether LDQ scores could be associated treatment outcomes in these patients. Third, these were self-administered instruments, and the reliability could be a concern. However, the responses to the questions in the instruments were checked at face-to-face interviews, and the findings should be reliable.

In conclusions, the presence of MOH in CM patients was associated with dependence behavior. Moreover, the LDQ was useful in the diagnosis or even screening of MOH in patients with CM. Further studies in independent clinical samples are needed.

## Figures and Tables

**Figure 1 diagnostics-13-00472-f001:**
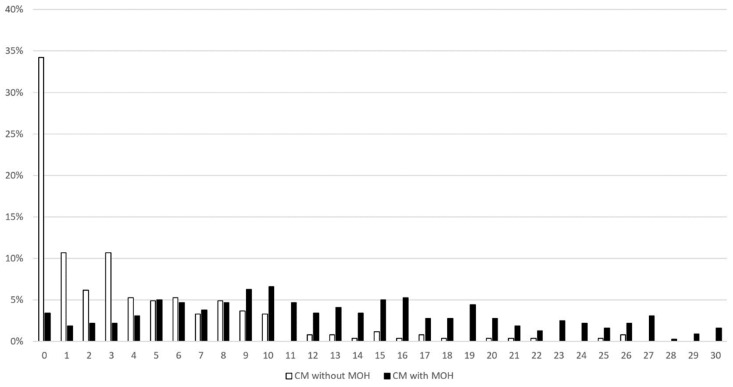
Distribution of LDQ scores in patients with and without MOH. Abbreviations: LDQ = Leeds Dependence Questionnaire, MOH = medication-overuse headache.

**Figure 2 diagnostics-13-00472-f002:**
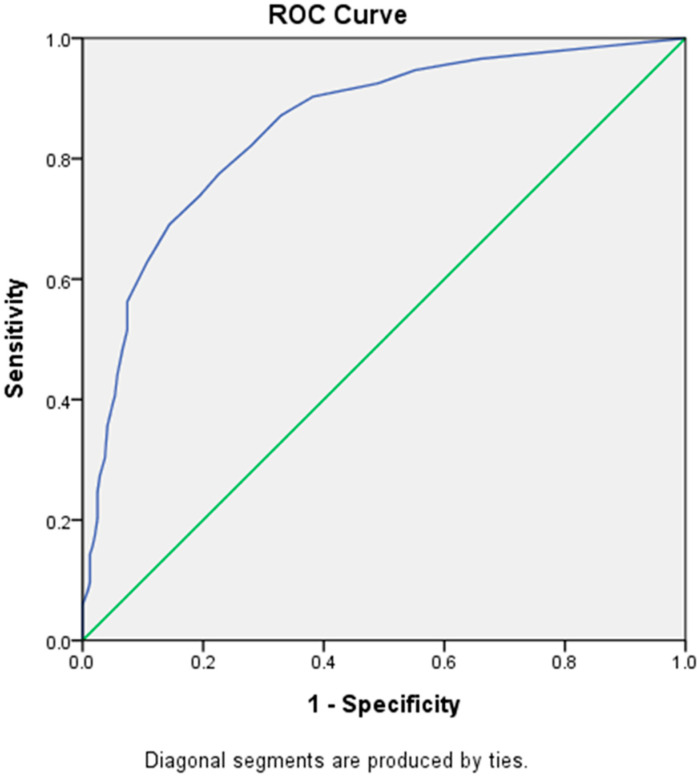
Receiver Operating Characteristic (ROC) curve of LDQ scores predictive of MOH. Abbreviations: LDQ = Leeds Dependence Questionnaire, MOH = medication-overuse headache.

**Figure 3 diagnostics-13-00472-f003:**
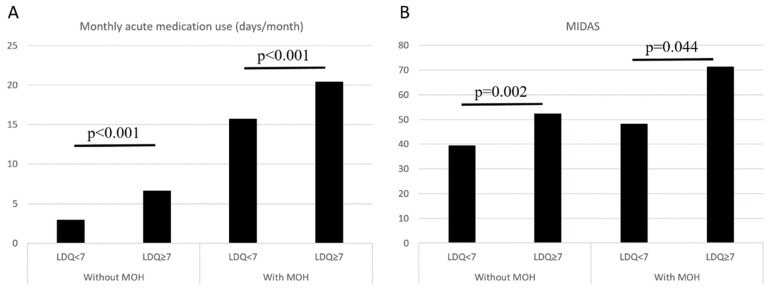
Comparisons on monthly acute medication use (**A**) and MIDAS scores (**B**) between patients with LDQ < 7 and LDQ ≥ 7 stratified by the presence of MOH. Abbreviations: LDQ = Leeds Dependence Questionnaire, MIDAS = Migraine Disability Assessment Scale, MOH = medication-overuse headache.

**Table 1 diagnostics-13-00472-t001:** Clinical characteristics in chronic migraine patients with and without medication-overuse headache.

	Without MOH (*n* = 243)	With MOH (*n* = 320)	*p* Value
Age	40.8 ± 12.5	42.3 ± 11.6	0.152
Female gender	197 (81.1%)	254 (79.4%)	0.618
Bachelor's degree or higher	136 (56.0%)	139 (43.4%)	0.003
Being married *	117 (48.1%)	163 (50.9%)	0.512
Being employed	157 (64.6%)	224 (70.0%)	0.176
Migraine with aura	13 (5.3%)	14 (4.4%)	0.592
Age at migraine onset (years)	22.8 ± 10.8	20.7 ± 9.2	0.065
Duration of CM (months)	60.4 ± 112.4	90.4 ± 100.6	<0.001
Average headache intensity on NRS (0–10)	6.2 ± 1.8	7.0 ± 1.9	<0.001
Monthly headache days	23.0 ± 6.9	23.8 ± 6.5	0.315
Monthly acute medication use (days/month)	3.7 ± 5.3	19.4 ± 7.8	<0.001
MIDAS (range 0–270)	42.5 ± 51.3	66.2 ± 77.4	<0.001
HADS-A (range 0–21)	9.7 ± 4.2	9.6 ± 4.3	0.621
HADS-D (range 0–21)	7.3 ± 4.1	8.0 ± 1.7	0.210
PSQI (range 0–21)	11.1 ± 4.2	11.9 ± 4.4	0.030

* Excluding those who were single, widowed, or divorced. Abbreviations: CM = chronic migraine, HADS = Hospital Anxiety and Depression Scale (A = anxiety subscale, D = depression subscale), MIDAS = Migraine Disability Assessment Scale, MOH = medication-overuse headache, NRS = numerical rating scale, PSQI = Pittsburgh Sleep Quality Index.

**Table 2 diagnostics-13-00472-t002:** Dependence behaviors in patients with and without medication-overuse headache (MOH).

Modified Leeds Dependence Questionnaire	Without MOH	With MOH	*p* Value
1. Do you find yourself thinking about when you will next be able to take analgesics?	0.5 ± 0.8	1.3 ± 1.1	<0.001
2. Is taking analgesics more important than anything else you might do during the day?	0.4 ± 0.7	1.5 ± 1.1	<0.001
3. Do you feel your need for analgesics is too strong to control?	0.4 ± 0.7	1.6 ± 1.1	<0.001
4. Do you plan your days around taking analgesics?	0.2 ± 0.5	1.0 ± 1.1	<0.001
5. Do you take analgesic in a particular way in order to increase the effect it gives you?	0.2 ± 0.6	0.7 ± 1.0	<0.001
6. Do you take analgesics morning, afternoon and evening?	0.3 ± 0.5	1.3 ± 1.0	<0.001
7. Do you feel you have to carry on taking analgesics once you have started?	0.5 ± 0.8	1.3 ± 1.1	<0.001
8. Is getting the effect you want more important than the particular analgesic you use?	0.6 ± 0.9	1.6 ± 1.1	<0.001
9. Do you want to take more analgesics when the effect starts to wear off?	0.3 ± 0.6	1.1 ± 1.0	<0.001
10. Do you find it difficult to cope with life without analgesics?	0.5 ± 0.8	1.7 ± 1.1	<0.001
Total score (range 0–30)	3.9 ± 5.1	13.0 ± 7.6	<0.001

## Data Availability

Data are available from the authors on reasonable requests.

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
