# Peer review of "Clinical Utility of Leeds Dependence Questionnaire in Medication-Overuse Headache"

_diagnostics, 2023, doi:10.3390/diagnostics13030472_

Round 1
Reviewer 1 Report
Please add a sentence about the internal and external validity of this study, where the limitations of the study have been explained.
Please add the application of the study findings in the clinic. Is that mainly for diagnostic purposes or can it be used for other purposes, for example, choice of treatment, de-prescription, or prescription of medications?
Author Response
Point #1. Please add the application of the study findings in the clinic. Is that mainly for diagnostic purposes or can it be used for other purposes, for example, choice of treatment, de-prescription, or prescription of medications?
Ans. Followed. Please refer to the last five lines of the first paragraph of Discussion.
Reviewer 2 Report
The presented paper is about dependence behaviors in medication-overuse headache (MOH) and the use of Leeds dependence questionnaire (LDQ) in this condition.
The study has good methodology, big study sample consisting of comparable patients. The manuscript is clear, relevant and well-structured. Most cited references are around 10 years old, just a minority are within the last 5 years. I did not detect excessive self-citations. The manuscript is scientifically sound and the design is appropriate. The results seem reproducible based on the given details. The figures and tables are appropriate, maybe the modified LDQ could be presented as a separate table, not as a paragraph of the text. Othewise the graphs and tables show the data properly and are sufficiently easy to understand. The conclusions of the study are consistent. The ethics statements and data availability statements are adequate.
The question of paper is well-defined and sufficiently novel as to my knowledge. The results provide some advance of the current knowledge. The work fits the journal scope. The results are appropriately interpreted and are significant, the conclusions are justified and supported by results. The hypotheses may be identified more specifically, characterisation is rather general. The article is written in an appropriate way. The data and analyses are presented properly. The study is correctly designed and technically sound. The data is robust enough to draw conclusions. The conclusions may be interesting for readership of the journal, but rather to headache specialist in centers, the questionnaire is probably not routinely used by primary care doctors. The work advances current knowledge somewhat.
The English language is appropriate and understandable with small number of grammar mistakes (e.g. a missing preposition).
I recommend to accept the paper after minor revisions – check the spelling and other grammar.
Consider presenting the Modified Leeds dependence questionnaire as a separate Table, not a paragraph in the text.
Author Response
The presented paper is about dependence behaviors in medication-overuse headache (MOH) and the use of Leeds dependence questionnaire (LDQ) in this condition.
Point #1. The study has good methodology, big study sample consisting of comparable patients. The manuscript is clear, relevant and well-structured. Most cited references are around 10 years old, just a minority are within the last 5 years. I did not detect excessive self-citations. The manuscript is scientifically sound and the design is appropriate. The results seem reproducible based on the given details. The figures and tables are appropriate, maybe the modified LDQ could be presented as a separate table, not as a paragraph of the text. Othewise the graphs and tables show the data properly and are sufficiently easy to understand. The conclusions of the study are consistent. The ethics statements and data availability statements are adequate.
Ans. Thanks for the positive comments. The reviewer’s point is well taken. The contents of modified LDQ is provided in table 2. Please refer to revised table 2.
Point #2. The question of paper is well-defined and sufficiently novel as to my knowledge. The results provide some advance of the current knowledge. The work fits the journal scope. The results are appropriately interpreted and are significant, the conclusions are justified and supported by results. The hypotheses may be identified more specifically, characterisation is rather general. The article is written in an appropriate way. The data and analyses are presented properly. The study is correctly designed and technically sound. The data is robust enough to draw conclusions. The conclusions may be interesting for readership of the journal, but rather to headache specialist in centers, the questionnaire is probably not routinely used by primary care doctors. The work advances current knowledge somewhat.
Ans. Thanks for the positive comments. The reviewer’s point is well taken. The hypothesis is elaborated. Please refer to the last four lines of the second paragraph of Introduction.
Point #3. The English language is appropriate and understandable with small number of grammar mistakes (e.g. a missing preposition).
I recommend to accept the paper after minor revisions – check the spelling and other grammar.
Ans. Thanks for the valuable suggestion. The manuscript was checked for spelling and grammatical errors, which are corrected in the revised manuscript. Please refer to lines 9-11 and 15 of abstract; line 8 of the first paragraph and lines 2-3 of the last paragraph of Introduction; lines, 3, 5-6, and 8 of the second paragraph, and lines 7-9 and 14 of the fourth paragraph of Methods; and the second to the last line of the first paragraph, line 5 of the second paragraph, and lines 1 and 4 of the third paragraph of Results; line 5 of the first paragraph, line 3, 8-9, 19 and 21 of the third paragraph, lines 2, 7, and 13 of the fourth paragraph, and line 2 of the last paragraph of Discussion.
Point #4. Consider presenting the Modified Leeds dependence questionnaire as a separate Table, not a paragraph in the text.
Ans. The reviewer’s point is well taken. The contents of modified LDQ is provided in table 2. Please refer to revised table 2.